# Timeline and Boundary Guided Diffusion Network for Video Shadow Detection

Haipeng Zhou
The Hong Kong University of Science and Technology (Guangzhou)
Guangzhou, China
hzhou321@connect.hkust-gz.edu.cn

Honqiu Wang
The Hong Kong University of Science and Technology (Guangzhou)
Guangzhou, China
hwang007@connect.hkust-gz.edu.cn

Tian Ye
The Hong Kong University of Science and Technology (Guangzhou)
Guangzhou, China
owentianye@hkust-gz.edu.cn

Zhaohu Xing
The Hong Kong University of Science and Technology (Guangzhou)
Guangzhou, China
zxing565@hkust-gz.edu.cn

Jun Ma
The Hong Kong University of Science and Technology (Guangzhou) & The Hong Kong University of Science and Technology
Guangzhou & Hong Kong SAR, China
eejma@hkust-gz.edu.cn

Ping Li
The Hong Kong Polytechnic University
Hong Kong SAR, China
p.li@polyu.edu.hk

Qiong Wang
Shenzhen Institute of Advanced Technology
Shenzhen, China
wangqiong@siat.ac.cn

Lei Zhu*
The Hong Kong University of Science and Technology (Guangzhou) & The Hong Kong University of Science and Technology
Guangzhou & Hong Kong SAR, China
leizhu@ust.hk

## ABSTRACT

Video Shadow Detection (VSD) aims to detect the shadow masks with frame sequence. Existing works suffer from inefficient temporal learning. Moreover, few works address the VSD problem by considering the characteristic (*i.e.,* boundary) of shadow. Motivated by this, we propose a Timeline and Boundary Guided Diffusion (TBGDiff) network for VSD where we take account of the past-future temporal guidance and boundary information jointly. In detail, we design a Dual Scale Aggregation (DSA) module for better temporal understanding by rethinking the affinity of the long-term and short-term frames for the clipped video. Next, we introduce Shadow Boundary Aware Attention (SBAA) to utilize the edge contexts for capturing the characteristics of shadows. Moreover, we are the first to introduce the Diffusion model for VSD in which we explore a Space-Time Encoded Embedding (STEE) to inject the temporal guidance for Diffusion to conduct shadow detection. Benefiting from these designs, our model can not only capture the temporal information but also the shadow property. Extensive experiments show that the performance of our approach overtakes the *state-of-the-art* methods, verifying the effectiveness of our components. We release the codes at https://github.com/haipengzhou856/TBGDiff.

## CCS CONCEPTS

• **Computing methodologies → Video segmentation**.

## KEYWORDS

Diffusion Model, Temporal Guidance, Boundary Attention, Video Shadow Detection

**ACM Reference Format:**
Haipeng Zhou, Honqiu Wang, Tian Ye, Zhaohu Xing, Jun Ma, Ping Li, Qiong Wang, and Lei Zhu. 2024. Timeline and Boundary Guided Diffusion Network for Video Shadow Detection. In *Proceedings of the 32nd ACM International Conference on Multimedia (MM '24), October 28-November 1, 2024, Melbourne, VIC, Australia.* ACM, New York, NY, USA, 10 pages. https://doi.org/10.1145/3664647.3681236

## 1 INTRODUCTION

Shadow detection is increasingly important in vision analysis and applications. Vision tasks can suffer from the shadows, including incorrect segmentation [52], inaccurate object detection [15, 25], and flawed tracking [10, 37]. Hence, shadow detection has become a highly focused area of research. Recently, one can witness significant progress in single Image Shadow Detection (ISD) [11, 25, 26, 79, 80], whereas in the dynamic scenario, Video Shadow Detection (VSD) is much more challenging.

Temporal correspondence information matters in video shadow detection. For example, the SC-Cor [16] focuses on the relationship between shadow and optical-flow. It relies on a contrastive loss to explore the temporal correspondence for adjacent frames. However,

*Corresponding author.

like other Unsupervised Video Object Segmentation (UVOS) [41, 43, 50, 56, 63, 73] methods, it still depends on additional clues (*i.e.*, optical-flow) and lack of semantic correspondence. We point out that the adjacent frames usually change slightly, leading the model to focus on the consistent area while distracting on the deformation region, which is more crucial for shadow detection.

In addition, few works notice the characteristic of shadow to propose a task-specific model. We are attracted to recent shadow removal works [19, 38], which use the boundary information to guide the restoration model to remove shadows. This indicates the boundary can provide potential clues to identify the shadows. Meanwhile, the contexts of boundary often contain higher uncertainty [59, 76, 78] making it difficult for the model to perform accurate segmentation. These observations motivate us to explore the boundary information in VSD. Moreover, the shadow detection works are dominated by CNNs [10, 11, 14, 16, 25, 26, 35, 79, 80] and Transformers [32, 34, 60, 69]. Advanced architectures can improve performance. For instance, Scotch&SODA [32], equipped with a Transformer-based backbone, surpasses other cutting-edge VSD methods by a large margin. Recently, Diffusion models present remarkable results in image generation [1, 4, 24, 44, 48] and semantic segmentation [2, 7, 9, 27]. Whereas little effort has been made to explore the temporal guidance of Diffusion model for Video. In the scheme of conditional Diffusion, guidance matters since it can instruct Diffusion to approach a distribution in a specific direction. Therefore, it is worth studying effective temporal guidance for Diffusion models to enhance the performance in VSD.

To tackle the aforementioned problems, we propose Timeline and Boundary Guided Diffusion (TBGDiff) for video shadow detection. ***To our best knowledge, this is the first work introducing Diffusion model for shadow detection.*** The core idea of our method is to utilize the temporal information in the clipped video and explore the boundary contexts for the Diffusion network to conduct VSD. In detail, (1) we design a Dual Scale Aggregation (DSA) module which is plug-and-play to aggregate the temporal features. Inspired by the residual operation in ResNet [22], we rethink the affinity [13, 39] in video condition. We adopt the vanilla affinity to capture the consistent context for short-term frames and propose a residual affinity to encourage the model to focus on the deformation area of shadows for long-term frames as well. (2) We present a Shadow Boundary-Aware Attention (SBAA) to encourage the model to represent the characteristics of shadows. We embed the boundary position into the attention mechanism [51] to guide the model to more accurately distinguish between shadow and non-shadow areas. (3) We explore three different temporal guidance for Diffusion model to detect shadows in video scenario. Via considering the timeline frames (*i.e.*, the past and future frames), we develop the best practice called Space-Time Encoded Embedding (STEE) to inject the temporal guidance into the conditional Diffusion model. Instead of using heavy U-Net [45] to predict noise, our TBGDiff can progressively decode the mask via the reverse process.

In summary, our four-fold contributions are:

- We develop Timeline and Boundary Guided Diffusion (TBGD -iff) for video shadow detection, which is the first work introducing Diffusion model to conduct shadow detection. Our TBGDiff outperforms *state-of-the-art* methods by a large margin, verifying the effectiveness of our approach.

- To guide the Diffusion to learn temporal information, we propose three different ways to produce guidance. The devised Space-Time Encoded Embedding (STEE) enables our model to capture the representation from a timeline sequence (past and future frames), resulting in the best performance.
- We develop a Shadow Boundary-Aware Attention to help the model understand the boundary context. Our model can be further improved by benefiting from focusing on the boundary-aware region.
- We introduce a plug-and-play method for video understanding, Dual Scale Aggregation (DSA), to explore the affinity in video sequence. Our DSA is able to conduct short-term consistency context learning and long-term visiting.

## 2 RELATED WORKS

### 2.1 Video Object Segmentation

Different from Semi-Supervised Video Object Segmentation (SSVOS) [12, 13, 29, 30, 39, 54] which provides the first frame mask to initialization during the testing stage, the Video Shadow Detection (VSD) follows the paradigm of Unsupervised Video Object Segmentation (UVOS) [28, 36, 41, 47, 50, 63] where we detect the shadow without the ground truth in the initial frame. In VOS, it usually deploys auxiliary encoder to extract temporal information, like optical flow [41, 43, 50, 63], vanilla encoder [36, 55, 57], or salient map [28, 47]. Recent studies [12, 13, 39, 54, 75] focus on the affinity between the past and current frames such that one can build a memory-bank, which aims to utilize the sequential information and past prior for video understanding. Not only in the past, but we also reconsider the information of future frames, *i.e.*, in a timeline sequence. We rethink the affinity in dual temporal scales to consider timeline aggregation instead of a sequential way.

### 2.2 Shadow Detection

Previous shadow detection works [11, 20, 25, 58, 79, 80] focus on the image shadow detection (ISD). For example, Zhu *et al.* [80] design a bidirectional FPN [31] to extract local and global contexts for detecting shadow. With respect to the dynamic scenarios, *i.e.*, video shadow detection, it encounters huger challenges on account of the complexity of the real-world. Chen *et al.* [10] collect the first video shadow detection dataset named ViSha, and introduce TVSD-Net to apply collaborative training in different videos to learn the shadow context. Similarly, STICT [35] deploys Teacher-Student model [23] to achieve video consistency learning. Considering the temporal correspondence, SC-Cor [16] enables the network to focus on the anchor pixel of shadow via contrastive learning. In order to balance temporal learning and contrastive learning, in Scotch&SODA [32] the authors apply trajectory Transformer to conduct VSD and set a new record on ViSha dataset. Differently, motivated by the shadow removal works [19, 38, 68] which utilizes the boundary information to guide the image restoration, we first design a specific shadow boundary-aware attention to detect the characteristic of shadows.

### 2.3 Diffusion Model

Diffusion model has shown remarkable promise in visual generation [4, 24, 42, 44, 48], and it enlightens other tasks like object detection [6], segmentation [2, 7, 9, 27] classification [21, 71]. For segmentation tasks, the denoise process is not suitable [8, 27] since the discrete signals undermine the segmentation. Bit Diffusion [7]

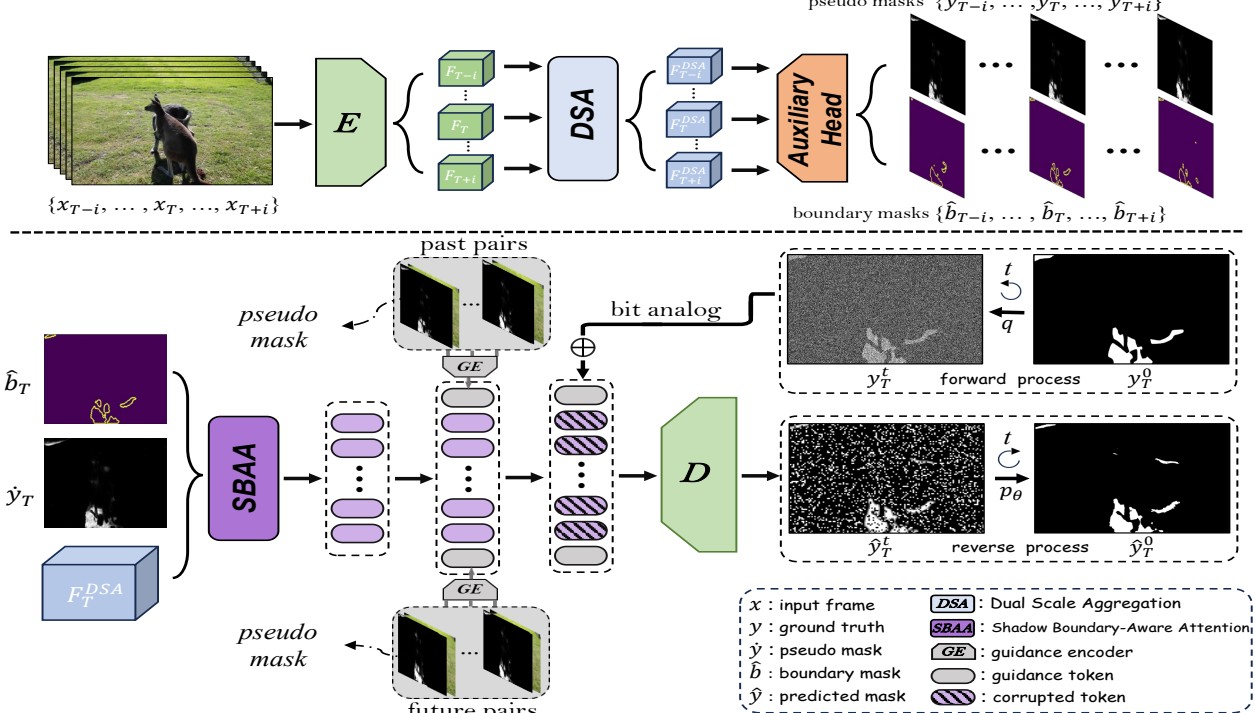

**Figure 1: Workflow of our $TBGDiff$. We first use an Encoder $E$ to represent all the frames, then the yielded features are sent to $DSA$ module to aggregate temporal features. The outputs of $DSA$ can be decoded as pseudo masks and boundary masks via an *Auxiliary Head*. For a frame from the sequence, we use $SBAA$ to further explore the shadow boundary context with given the boundary mask $\hat{b}_T$, pseudo mask $\dot{y}_T$, and aggregated feature $F_T^{DSA}$. Such that, the tokens produced by $SBAA$ and timeline guidance generated by $GE$ can be used for Diffusion to conduct video shadow detection.**

introduces a simple and generic way to embed and analog the discrete mask into continuous signals, and it also presents a simple concatenation strategy to process VOS. While there is still room to improve the temporal guidance for Diffusion to tackle VOS. Moreover, the efficiency should be considered as well. Though several works [4, 48, 49] dedicate to accelerate the inference, the conventional Diffusion models [1, 4, 24, 40, 44, 48, 74, 77] use heavy U-Net for noise estimation. Huge parameters make it hard to go on the downstream works. Instead, we further explore the feasibility and temporal understanding of the Diffusion model for VSD.

## 3 METHODOLOGY

### 3.1 Overview

Given $2i + 1$ frames, our TBGDiff can simultaneously detect the shadow for all clipped sequences. In brevity, we illustrate the workflow of the $T$-th frame $x_T$ in Fig. 1. First, we take an encoder to extract the temporal-agnostic features for all the frames, then the features are sent to the Dual Scale Aggregation (DSA) module to implement temporal aggregation. With an Auxiliary Head, these aggregated features are used to produce pseudo masks and boundary masks. Next, we input the $T$-th aggregated feature $F_T^{DSA}$, pseudo mask $\dot{y}_T$, and boundary $\hat{b}_T$ to Shadow Boundary-Aware Attention (SBAA) to further explore the characteristic of shadows. To utilize the timeline temporal information for Diffusion, we use a guidance encoder to yield Space-Time Encoded Embedding (STEE) via encoding the past and future pairs (pseudo masks and images). Finally,

we adopt bit analog strategy [7] to embed noise and conduct the denoise process to predict the final shadow masks.

### 3.2 Dual Scale Aggregation

When it comes to video-related works, the matching-based methods [13, 18, 39, 61, 62, 70, 72, 73, 75] usually adopt affinity to read and visit the space-time correspondences. However, the vanilla affinity will introduce the concern: it intends to give more weight to the adjacent frames because the contexts of close-range sequences change smoothly and slightly, and the interval frames gain less attention due to time-shift. We argue that both of the temporal scales of features should be considered, and to alleviate the temporal bias we propose a Dual Scale Aggregation (DSA) module where we rethink the affinity considering short-term and long-term scenarios jointly.

The core of affinity is to compute the similarity between the query feature and the memory feature to retrieve the temporal and spatial feature. Given the query feature $Q \in \mathbb{R}^{C_Q \times HW}$, the memory key feature $K \in \mathbb{R}^{C_K \times NHW}$, and memory value feature $V \in \mathbb{R}^{C_V \times NHW}$ (the $H$ and $W$ are the spatial dimensions, the $C_{\{Q,K,V\}}$ denote the channels, and $N$ represents the memory length), the affinity $M \in \mathbb{R}^{NHW \times HW}$ is computed as:

$$M_{(a,b)}(Q, K) = \frac{exp(f(Q_{(a)}, K_{(b)}))}{\sum_x exp(f(Q_{(x)}, K_{(b)}))}, \quad (1)$$

where $f(\cdot)$ is L2 similarity function [13]. Such that, we can readout the aggregated feature $F \in \mathbb{R}^{C_V \times HW}$ via the matrix multiplication:

$$F = VM(Q, K). \quad (2)$$

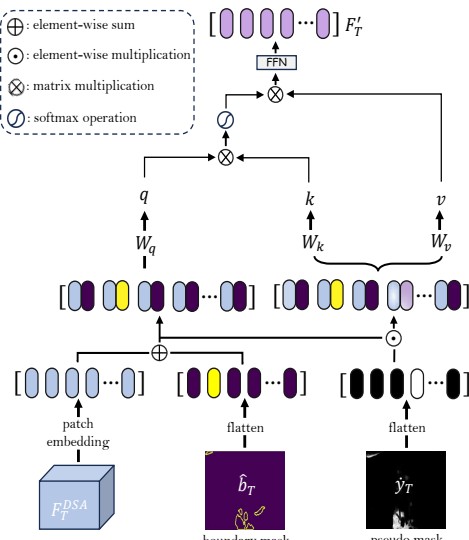

**Figure 2: Illustration of our SBAA. By integrating the $\hat{b}_T$ and $F_T^{DSA}$, we can obtain the boundary-aware embedded tokens serving as the *query*. We also use the pseudo mask to weight the coarse shadow regions via element-wise multiplying these tokens to produce *key* and *value*. Such that, we can implement attention mechanism and FFN to output the boundary-aware and shadow-oriented features.**

Such a vanilla affinity can be deployed for short-term aggregation. With the current frame feature $F_T$ as $Q$, and the concatenated adjacent features $\{F_{T-1}, F_{T+1}\}$ as the $K^s$ and $V^s$ (see Fig. 1), we readout the short-term aggregated feature via the vanilla affinity:

$$F_T^s = V^s M^s(Q, K^s). \qquad (3)$$

Considering the long-term visiting, Eq. 1 indicates that similar areas among different frames occupy higher weight. This leads the model to distract from the deformation region. To encourage the network to focus on it, we propose residual affinity to enhance the weight of deformation areas. Similarly, we have the query of $F_T$, the key and value $K^l = V^l = \{F^l\}$ where $l \in [T-i, ..., T-2, T+2, ..., T+i]$, and the long-term aggregated feature can be obtained by a residual operation with the affinity matrix:

$$F_T^l = V^l M^l(Q, K^l) = V^l \left( \left| M^{self}(Q, Q') - M^l(Q, K^l) \right| \right), \qquad (4)$$

where $M^{self}$ is a self-affinity anchored the consistent areas and the $Q'$ is the broadcasting version of $Q$ to match the size. An explanation is that the close-range frames usually contain consistent area leading to higher similarity, while the discrepancy area receives less attention. Adopting a subtraction operation on the affinity matrix, the residual area will reveal the difference region which is crucial to track the shadow deformation. With $F_T^s$ and $F_T^l$, our DSA can produce the dual scales aggregated features for $T$-th frame via a simple convolutional residual block:

$$F_T^{DSA} = ResBlock(F_T, F_T^s, F_T^l). \qquad (5)$$

### 3.3 Shadow Boundary-Aware Attention

Considering the property of shadows, previous shadow removal works [19, 38, 68] suggest that the marginal contexts of shadows indicate crucial clues to identify the shadow and non-shadow regions.

Motivated by this, we design a Shadow Boundary-Aware Attention (SBAA) for specializing in detecting the shadows.

First, we introduce an Auxiliary Head in which we input the aggregated features $F^{DSA}$ to produce the pseudo mask $\dot{y}$ and boundary mask $\hat{b}$. Benefiting from this design, the pseudo mask can produce the semantic guidance for our Diffusion (see Sec. 3.4), and the boundary mask helps the model capture the characteristics of shadows. Here, we use the auxiliary loss to supervise the production via:

$$L_{aux} = L_{bce}(b, \hat{b}) + L_{bce}(y, \dot{y}), \qquad (6)$$

where $b$ and $y$ are the ground truths of the boundary and shadow masks, respectively. We adopt Binary Cross Entropy ($L_{bce}$) to compute loss. Note that all the frames are conducted.

Then move to a specific frame at $T$, given the aggregated feature $F_T^{DSA} \in \mathbb{R}^{C \times H \times W}$ from DSA, the predicted boundary $\hat{b}_T$, and the pseudo mask $\dot{y}_T$, we regard $\hat{b}_T$ as the position embedding such that the boundary-aware embedded tokens can be obtained by:

$$F_T' = [f_T^1 \mathbf{E}; f_T^2 \mathbf{E}; ...; f_T^n \mathbf{E}] + \mathbf{E}_{bp}, \quad n = H \times W, f_T^i \in F_T^{DSA}, \quad (7)$$

where $\mathbf{E}$ is a patch embedding projection [17], and we flatten the boundary mask to obtain $\mathbf{E}_{bp}$ serving as the boundary-aware position embedding. To highlight the shadow region, we propose the SBAA which can be described as:

$$q = F_T' W_q, \quad k = (F_T' \cdot \dot{y}_T) W_k, \quad v = (F_T' \cdot \dot{y}_T) W_v, \qquad (8)$$

$$SBAA = Softmax(\frac{qk^{tr}}{\sqrt{C}})v, \qquad (9)$$

where $W_q$, $W_k$, and $W_v$ are the learnable matrices, and $C$ is the number of channel to scale. We implement broadcasting mechanism to extend the channels of $\dot{y}_T$ to match the size of $F_T'$, and $\dot{y}_T$ can serve as the weights of probability to emphasize the shadow-relevant areas. A visual depiction of our SBAA can be found in Fig. 2.

By doing so, the boundary-aware query is able to better retrieve the shadow regions. And based on the attention, we can deploy feed-forward network (FFN) [51] (*i.e.*, the MLP linear layer) on it to produce the output feature which is used in our Diffusion process to give the final prediction.

### 3.4 Space-Time Encoded Embedding Guidance

Recently, Diffusion Models have indicated powerful ability in segmentation tasks [2, 7, 27, 65]. When it comes to VOS, Pix2Seq [7] adopts a straightforward way that just concatenates the past predicated masks into Diffusion as guidance to conduct VOS. While, the potential of conditional Diffusion in VOS has yet to be further explored. Motivated by this, we try to seek more effective temporal guidance for Diffusion models and deploy it in VSD.

Briefly, Diffusion model [24, 48] contains a forward process $q$ and a reverse process $p$. The $q$ aims to gradually add Gaussian noise to corrupt the distribution of a mask $y^0$ making it close to a normal distribution, which can be illustrated as:

$$q(y^t|y^0) = \sqrt{\bar{\alpha}^t} y^0 + \sqrt{(1-\bar{\alpha}^t)} \epsilon, \quad \epsilon \sim \mathcal{N}(0, 1). \qquad (10)$$

The $t$ denotes the timestep, and $\bar{\alpha}^t$ is a noise scheduler which could be adjusted by a cosine or linear style. And the reverse process $p_\theta$ is parameterized by a network $\theta(y^t, g, x)$, which is used to predict $y^0$

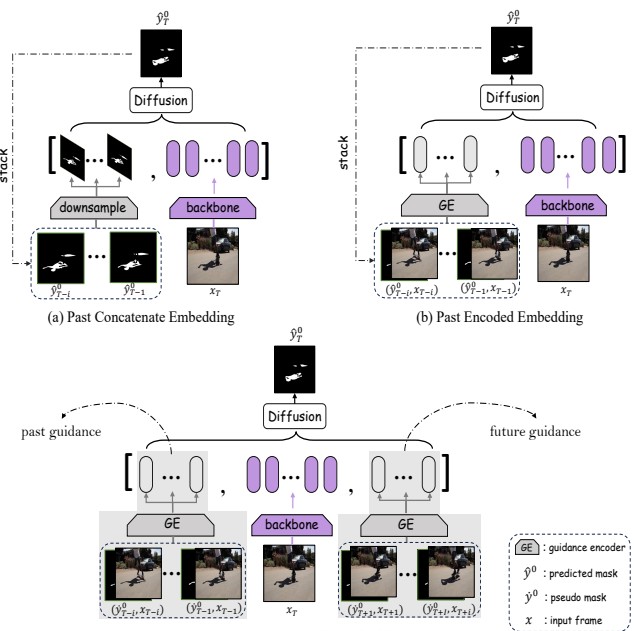

(a) Past Concatenate Embedding        (b) Past Encoded Embedding

(c) Space-Time Encoded Embedding

**Figure 3: Three different ways to produce guidance for conditional Diffusion. (a) PCE simply concatenates the predicted masks to current features as the temporal guidance. (b) PEE adopts the past encoded embedding as guidance which is more robust. (c) STEE encodes the pseudo masks and image pairs in both past and future to guide the Diffusion.**

from $y^t$ step by step based on the condition guidance $g$ and image $x$. This Markov process can be written as:

$$p_\theta(y^{0:t}|g,x) = p(y^t) \prod p_\theta(y^{t-1}|y^t,g,x). \tag{11}$$

Instead of predicting noise like conventional Diffusion models, we predict the mask since the robust representation and bit analog strategy [8, 27] controlled by a scale weight enable the model to directly decode the mask rather than relying on the heavy U-Net. The detail of the Diffusion's operation are provided in ***Supplementary Material***, and the ablation studies of the hyperparameters (scale weight, noise scheduler, and sampling steps) are given later. Here, we mainly discuss the importance of the guidance $g$.

In the scheme of Diffusion, the conditional guidance is usually accessible (*e.g.,* vectors of text or images). Considering the VOS, Pix2Seq [7] introduces a loop to predict the masks frame by frame such that the obtained predictions can serve as a temporal guidance to promote the following segmentation. It simply downsamples the masks and concatenates them with the latent features, and we denote it as Past Concatenate Embedding (PCE, see Fig. 3(a)). While, existing Latent Diffusion models [4, 44] have proved that in latent space the Diffusion can perform better, which suggests us rethink the embedding ways. Intuitively, we propose other two methods to embed the guidance: Past Encoded Embedding (PEE, see Fig. 3(b)) and Space-Time Encoded Embedding (STEE, see Fig. 3(c)). We concatenate the masks with corresponding frames to form pairs, then a light-weight guidance encoder is deployed to embed them. Compared to PCE, the encoded guidances from PEE and STEE are much more robust to visit the temporal information.

Here, we point out that the unidirectional embeddings (PCE & PEE) are not the best solution. They bring the following concerns: (1) The PCE and PEE access the guidance sequentially, leading to lower efficiency. Since the embedding is conducted frame by frame, they are restricted to real-time online shadow detection. Moreover, the uncertainty can accumulate as well. (2) The future prediction is agnostic, resulting in limited temporal guidance usage. All the space-time clipped frames should be considered to provide a temporal context. To address the aforementioned issues, we devise STEE to use all the space-time information in an efficient way. Instead of using the predicted masks, we take use of the pseudo masks produced by an Auxiliary Head (see Fig. 1 and Sec. 3.3). Because all the pseudo masks are available, we can execute the guidance encoding in a parallel manner rather than wait for the last prediction. As a result, our Diffusion can visit all the timeline temporal information leading to better performance.

### 3.5 Objective Loss
As mentioned in the last section, we directly predict the masks rather than the noise. Hence, the objective loss is similar to the segmentation task. Here, we adopt the Binary Cross Entropy [66, 67] and lovasz-hinge loss [3] to restrict training. Considering the auxiliary loss, the total term of our loss function is computed as:

$$\mathcal{L}_{seg} = L_{bce} + L_{hinge} + L_{aux}. \tag{12}$$

## 4 EXPERIMENT
We use Video Shadow dataset (ViSha) [10] to conduct our experiments and make comparisons with *state-of-the-art* methods. The ViSha dataset contains 50 videos for training and 70 videos for testing. Following previous studies [10, 16, 32, 35, 60], we deploy Mean Absolute Error (MAE), Intersection over Union (IoU), F-measure score ($F_\beta$), and Balance Error Rate (BER) as evaluation metrics for quantitative comparisons. Regarding the BER, we also compute the S-BER score at the shadow regions and the N-BER score at non-shadow regions, respectively.

### 4.1 Implement details
We utilize AdamW optimizer [33] with a learning rate of 3e-5 to train our model. The batch size is 4, and the clipped sequence is 5 frames. Four A6000 GPUs are used to conduct our experiments, and a fixed random seed ensures the reproduction. For a fair comparison, following Scotch&SODA [32], we use MiT-B3 [64] as our feature extraction backbone, and all experiments are conducted with a resolution of 512×512. Note that the boundary masks are obtained by utilizing the canny operator on the shadow masks. More setup details are provided in ***Supplementary Material***.

### 4.2 Comparisons with *State-of-the-art* Methods
*4.2.1 Compared Methods.* We compare our network against 20 cutting-edge methods, including Image Object Segmentation (IOS) [5, 27, 31, 64], Image Shadow Detection (ISD) [11, 14, 25, 69, 79, 80], Video Object Segmentation (VOS) [7, 13, 36, 39, 53], and Video Shadow Detection (VSD) [10, 16, 32, 35, 60]. Note that the Semi-Supervised video segmentation methods like STM [39], STCN [13], and ShadowSAM [60] are not given the label of the first frame during testing, we reproduce them by predicting the initial frame for fair comparisons in line with previous VSD methods and ours.

**Table 1: Quantitative comparisons with *state-of-the-art* methods. We compare with several methods from the Image Object Segmentation (IOS), Image Shadow Detection (ISD), Video Object Segmentation (VOS), and Video Shadow Detection (VSD). The † denotes the methods that require ground truth for initialization in testing, for fair comparisons we directly predict the first frame instead of using the label. Bold indicates the best performances, and underline indicates the second-best performances.**

| | METHODS | | METRICS | | | | | |
|---|---|---|---|---|---|---|---|---|
| Tasks | Models | Venues | MAE ↓ | $F_\beta$ ↑ | IoU ↑ | BER ↓ | S-BER ↓ | N-BER ↓ |
| IOS | FPN [31] | CVPR17 | 0.044 | 0.707 | 0.512 | 19.49 | 36.59 | 2.40 |
| | Deeplabv3+ [5] | ECCV18 | 0.049 | 0.695 | 0.475 | 17.86 | 33.77 | 1.98 |
| | Segformer [64] | NIPS21 | 0.030 | 0.773 | 0.601 | 11.56 | 21.39 | 1.73 |
| | DDP [27] | ICCV23 | 0.038 | 0.771 | 0.608 | 10.74 | 18.90 | 2.57 |
| ISD | BDRAR [80] | ECCV18 | 0.050 | 0.695 | 0.484 | 21.29 | 40.28 | 2.31 |
| | DSD [79] | CVPR19 | 0.043 | 0.702 | 0.518 | 19.88 | 37.89 | 1.88 |
| | MTMT [11] | CVPR20 | 0.043 | 0.729 | 0.517 | 20.28 | 38.71 | 1.86 |
| | FSDNet [25] | TIP21 | 0.057 | 0.671 | 0.486 | 20.57 | 38.06 | 3.06 |
| | SDDNet [14] | MM23 | 0.040 | 0.754 | 0.548 | 14.05 | 26.10 | 1.61 |
| | SILT [69] | ICCV23 | 0.031 | 0.796 | 0.606 | 12.80 | 24.29 | 1.29 |
| VOS | COSNet [36] | CVPR19 | 0.040 | 0.705 | 0.514 | 20.50 | 39.22 | 1.79 |
| | FEELVOS [53] | CVPR19 | 0.043 | 0.710 | 0.512 | 19.76 | 37.27 | 2.26 |
| | †STM [39] | ICCV19 | 0.064 | 0.639 | 0.447 | 23.77 | 43.88 | 3.65 |
| | †STCN [13] | NIPS21 | 0.048 | 0.684 | 0.528 | 12.42 | 21.36 | 3.48 |
| | Pix2Seq [7] | ICCV23 | 0.034 | 0.775 | 0.618 | 10.63 | 19.13 | 2.14 |
| VSD | TVSD [10] | CVPR21 | 0.033 | 0.757 | 0.567 | 17.70 | 33.97 | 1.45 |
| | STICT [35] | CVPR22 | 0.046 | 0.702 | 0.545 | 16.60 | 29.58 | 3.59 |
| | SC-Cor [16] | ECCV22 | 0.042 | 0.762 | 0.615 | 13.61 | 24.31 | 2.91 |
| | †ShadowSAM [60] | TCSVT23 | 0.034 | 0.754 | 0.575 | 12.58 | 23.60 | 1.57 |
| | Scotch&SODA [32] | CVPR23 | 0.029 | 0.793 | 0.640 | 9.07 | 16.26 | 1.44 |
| | Ours | / | **0.023** | **0.797** | **0.667** | **8.58** | **16.00** | **1.15** |

**Table 2: Comparisons on the model size and speed of our network and *state-of-the-art* video shadow detectors.**

| Methods | Params (MB) | FPS | IoU↑ | BER↓ |
|---|---|---|---|---|
| TVSD [10] | 243.3 | 3.56 | 0.567 | 17.70 |
| STICIT [35] | 104.7 | 8.54 | 0.545 | 16.60 |
| SC-Cor [16] | 232.6 | 5.44 | 0.615 | 13.61 |
| SCOTCH&SODA [32] | 211.8 | 12.60 | 0.640 | 9.07 |
| ShadowSAM [60] | **101.3** | 13.10 | 0.575 | 12.58 |
| **Ours (TBGDiff)** | 102.3 | **14.01** | **0.667** | **8.58** |

**Table 3: Ablation study on our different modules. Here, the DIFFUSION adopts STEE to obtain temporal guidance.**

| SETTING | DIFFUSION | SBAA | DSA | METRIC | | | |
|---|---|---|---|---|---|---|---|
| | | | | IoU ↑ | BER↓ | $F_\beta$ ↑ | MAE↓ |
| Baseline | ✔ | | | 0.636 | 9.42 | 0.779 | 0.028 |
| M1 | ✔ | ✔ | | 0.648 | 9.33 | 0.782 | 0.030 |
| M2 | ✔ | | ✔ | 0.654 | 8.72 | 0.783 | 0.024 |
| **Ours** | ✔ | ✔ | ✔ | **0.667** | **8.58** | **0.797** | **0.023** |

*4.2.2 Quantitative Comparisons.* We report the quantitative results of our TBGDiff and compared methods in Tab.1. Among the 20 compared methods, Scotch&SODA [32] has the best MAE score of 0.029, the best IoU score of 0.640, the best BER score of 9.07, and the best S-BER score of 16.26, while SILT [69] ranks the first place in terms of $F_\beta$ (0.796) and N-BER (1.29). Our method outperforms all of *state-of-the-art* methods considering all the metrics. In detail, our TBGDiff improves the MAE score from 0.029 to 0.023, the $F_\beta$ score from 0.029 to 0.023, the IoU score from 0.796 to 0.797, the BER score from 9.07 to 8.58, the S-BER score from 16.26 to 16.00, and the N-BER score from 1.29 to 1.15, respectively.

*4.2.3 Qualitative Comparisons.* We demonstrate the visual comparisons of video shadow detection results of our network and *state-of-the-art* methods in Fig. 4. It is obvious that our method can not only better localize shadow regions but also identify the shadow boundaries more accurately. In contrast, other methods

either miss some shadow pixels or detect many non-shadow areas in their results. For example, in the $1^{st}$ row of Fig. 4, other methods tend to wrongly identify the black headphone as shadows, while our method can alleviate such mistakes. In the $3^{rd}$ and $5^{th}$ rows of Fig. 4, compared methods fail to identify the complex shadow regions, while our network can better detect these complex shadows due to integrating the shadow boundaries. From these visual depictions, we can conclude that our approach provides an effective solution to address the challenging video shadow detection task. See more visual comparisons in our ***Supplementary Material***.

*4.2.4 Efficiency Comparisons.* We also compare the efficiency of our method with other video shadow detection works in Tab. 2. Though the parameters of our model (102.3MB) are slightly larger than the smallest one (101.3MB), our approach takes the first rank in terms of the FPS and performance metrics, indicating our solution is much more effective and efficient for video shadow detection.

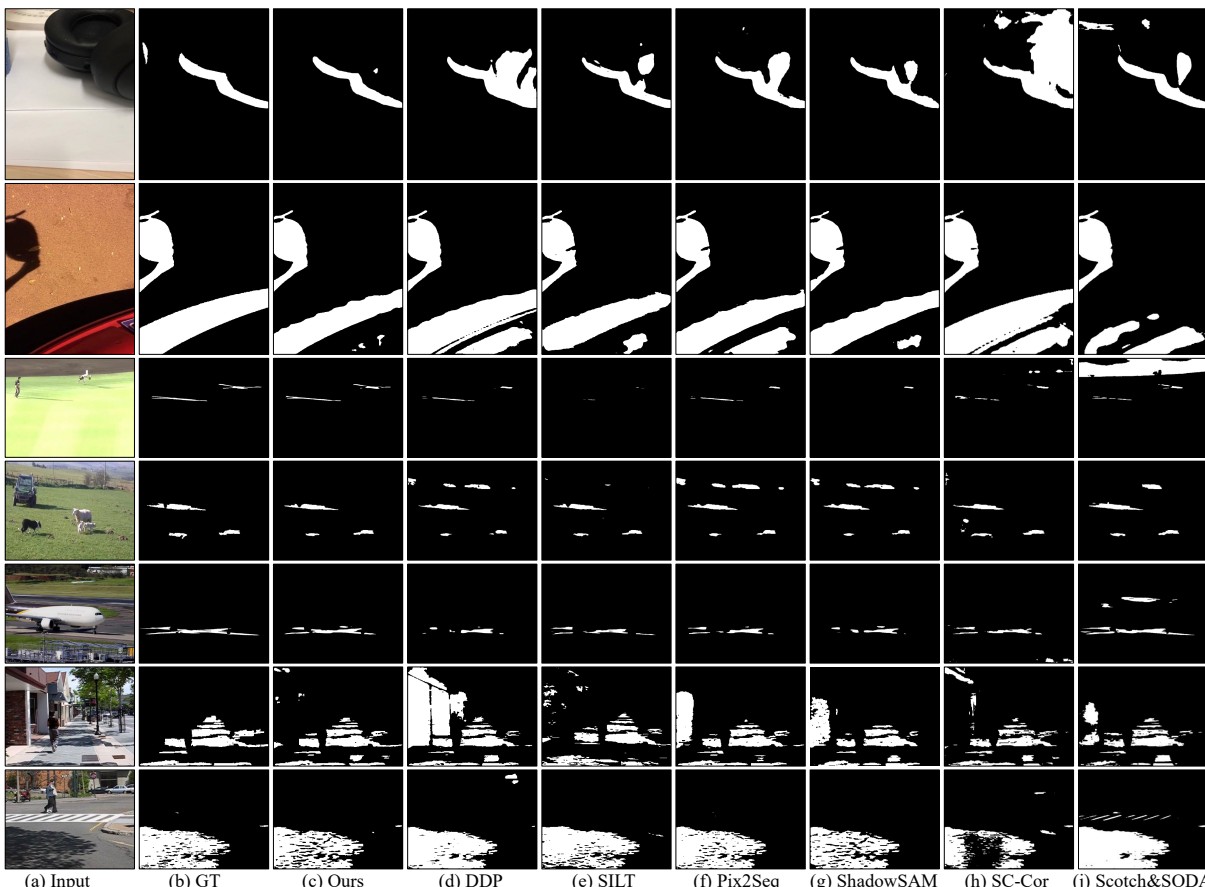

| (a) Input | (b) GT | (c) Ours | (d) DDP | (e) SILT | (f) Pix2Seq | (g) ShadowSAM | (h) SC-Cor | (j) Scotch&SODA |

**Figure 4: Visual comparisons with *state-of-the-art* methods. Apparently, our predicted masks show fewer noises and more accurate boundary correlation to shadows. See more compared results in our *Supplementary Material*.**

**Table 4: Ablation study on different temporal scales of DSA module. We conduct experiments on top of our TBGDiff.**

| SETTING | DSA | | METRIC | | | |
|---|---|---|---|---|---|---|
| | Short | Long | IoU ↑ | BER↓ | $F_\beta$ ↑ | MAE↓ |
| Ours w/o DSA (M1) | | | 0.648 | 9.33 | 0.782 | 0.030 |
| ① | ✔ | | 0.655 | 10.11 | 0.790 | 0.026 |
| ② | | ✔ | 0.650 | 9.00 | 0.779 | 0.026 |
| **Ours** | ✔ | ✔ | **0.667** | **8.58** | **0.797** | **0.023** |

**Table 5: Ablation study on different ways of producing temporal guidance for Diffusion model.**

| SETTING | GUIDANCE | | | METRIC | | | |
|---|---|---|---|---|---|---|---|
| | PCE | PEE | STEE | IoU ↑ | BER↓ | $F_\beta$ ↑ | MAE↓ |
| i | ✔ | | | 0.621 | 11.03 | 0.778 | 0.029 |
| ii | | ✔ | | 0.644 | 10.31 | 0.789 | 0.027 |
| **Ours** | | | ✔ | **0.667** | **8.58** | **0.797** | **0.023** |

quantitative results suggest our SBBA can effectively improve the performance when considering the boundary information.

*4.3.2 Effectiveness of DSA.* By observing Tab. 3, with given DSA module the configured models yield obvious improvements (Baseline&M2, and M1&Ours). It indicates that the DSA module also improves the video shadow detection performance of our network by aggregating temporal features from input video frames. In addition, we report the ablation studies on different scales of temporal aggregation in Tab. 4. From the results, it can be observed that each scale of aggregation can improve the shadow detection performance. By utilizing both long-term and short-term aggregation, our approach achieves the best results. We also visualize the readout results in Fig. 5, where we select the mid-frame (*i.e.*, the third one over five frames) for the best view. The long-term readout will focus on the deformable areas in the second column. Though it

## 4.3 Ablation Studies

We first conduct an ablation study to evaluate the effectiveness of our two modules (*i.e.*, SBBA, and DSA) on the Diffusion model. To do so, we build a "Baseline" by removing both SBBA and DSA modules from our network. Then "M1" and "M2" are reconstructed by adding the SBBA module and the DSA module into "Baseline". Tab. 3 reports the quantitative results of our network and three constructed networks ("Baseline", "M1", and "M2").

*4.3.1 Effectiveness of SBBA.* In Tab. 3, we can find that "M1" achieves an IoU improvement of +1.2%, a BER improvement of +9%, an $F_\beta$ improvement of 0.3%, and an MAE improvement of 0.2%, when compared to "Baseline". Moreover, "M2" encounters apparent degradation without using SBAA compared to our final models. These

**Table 6: Ablation study on the hyperparameters of Diffusion configuration, including the (a) noise scheduler, (b) the scale weight in bit analog strategy [7], and (c) the sampling steps in Diffusion.**

| (a) Noise Scheduler. Cosine does the best. | | | |
| --- | --- | --- | --- |
| Scheduler | IoU ↑ | BER ↓ | F$_\beta$ ↑ |
| **cosine (ours)** | **0.667** | **8.58** | **0.797** |
| linear | 0.641 | 9.87 | 0.777 |

| (b) Scale weight. The best factor is 0.01. | | | |
| --- | --- | --- | --- |
| Scale | IoU ↑ | BER ↓ | F$_\beta$ ↑ |
| 0.1 | 0.619 | 10.33 | 0.762 |
| **0.01 (ours)** | **0.667** | **8.58** | **0.797** |
| 0.001 | 0.633 | 9.94 | 0.754 |

| (c) Step. Sampling 20 steps is the best. | | | |
| --- | --- | --- | --- |
| Step | IoU ↑ | BER ↓ | F$_\beta$ ↑ |
| 10 | 0.639 | 9.40 | 0.779 |
| **20 (ours)** | **0.667** | **8.58** | **0.797** |
| 30 | 0.660 | 9.12 | 0.784 |

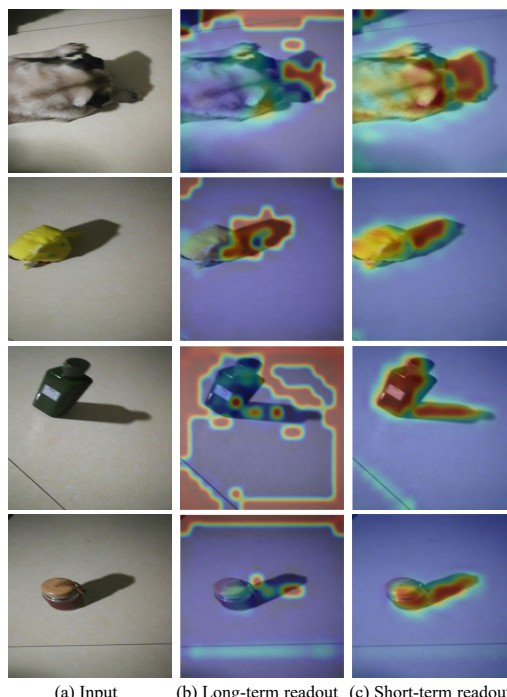

(a) Input    (b) Long-term readout    (c) Short-term readout

**Figure 5: Grad-CAM [46] visualization of the readout when conducting (b) long-term and (c) short-term aggregation**

will introduce noises, it can still concentrate on the shadow areas. As to the short-term, it encourages reading the consistent context and semantics to further enhance the understanding of shadows as shown in the third column.

*4.3.3 Discussion on the Diffusion Guidance.* As shown in Fig. 3, we design three different ways (*i.e.*, PCE, PEE, and STEE) to produce the temporal guidance for conditional Diffusion. Tab. 5 reports the quantitative results of our network with PCE, PEE, and STEE. Compared to PCE, PEE enables our network to achieve a better video shadow detection result. It indicates that taking features extracted from the past and predicted masks can work better as the Diffusion guidance when compared to simply concatenating them. By further incorporating the future frames guidance (*i.e.*, timeline), our network with STEE yield the best practice. It shows that guidance produced by timeline frames enables our network to achieve better results. Moreover, we provide the visual results with PCE, PEE, and STEE in Fig. 6, which further proves that our network with STEE has the best video shadow detection performance.

*4.3.4 Diffusion hyperparameters.* We also conduct the ablation studies to discuss the aforementioned hyperparameters of Diffusion

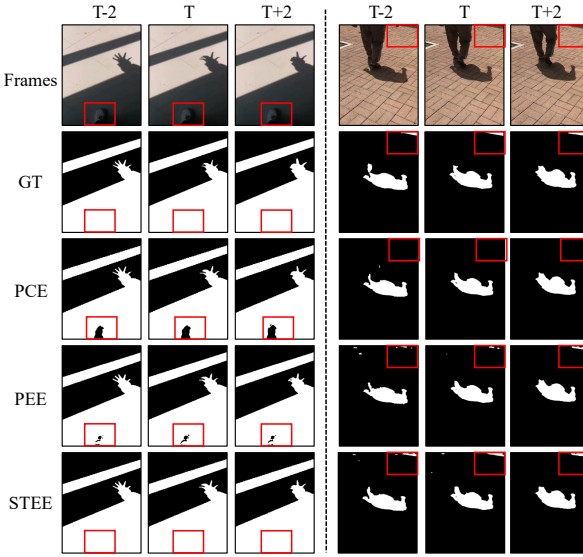

**Figure 6: The visual comparisons about Diffusion model guided by different ways. PCE intends to predict the darker area and lack of temporal perception. PEE can alleviate this problem by providing more robust guidance. Our STEE presents the best practice based on the timeline guidance.**

model (see Sec. 3.4), including the noise scheduler, scale weight, and the sampling step, and report the corresponding quantitative results in Tab. 6. According to the numerical results, we find that our network has the best video shadow detection performance when using a cosine scheduler, a scale weight of 0.01, and a sampling step of 20. Hence, we empirically adopt a cosine scheduler, 20 sampling steps, and a scale weight of 0.01 in our experiments.

## 5 CONCLUSION

In this work, we propose a Timeline and Boundary Guided Diffusion (TBGDiff) network which is the first work to use Diffusion model for video shadow detection task. The main idea of our TBGDiff is to extract temporal guidance for Diffusion and to utilize the boundary information to capture the characteristics of shadows. In detail, we propose a Dual Scale Aggregation (DSA) module to aggregate the temporal signals by rethinking the discrepancy of the affinity among short-term and long-term. We also devise an Auxiliary Head to yield boundary masks and pseudo masks, which can be used for extracting the boundary context of shadows by Shadow Boundary-Aware Attention (SBAA) and producing timeline temporal guidance via Space-Time Encoded Embedding (STEE) for Diffusion, respectively. Experimental results show that the developed designs are effective and our approach can outperform *state-of-the-art* methods.

**Acknowledgment.** This work is supported by the Guangzhou-HKUST(GZ) Joint Funding Program (No. 2023A03J0671), the InnoHK funding launched by Innovation and Technology Commission, Hong Kong SAR, the Guangzhou Industrial Information and Intelligent Key Laboratory Project (No. 2024A03J0628), the Nansha Key Area Science and Technology Project (No. 2023ZD003), and Guangzhou-HKUST(GZ) Joint Funding Program (No. 2024A03J0618).

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
