# OpenReview forum: "Timeline and Boundary Guided Diffusion Network for Video Shadow Detection"
_acmmm.org/ACMMM/2024/Conference — MM2024 Oral_

### Official Review · Reviewer_oqqD · 2024-05-20

**Rating:** 4
**Confidence:** 3

**Summary:**

The paper introduces a diffusion-based method for video shadow detection (VSD). In this work, a timeline and boundary guided diffusion network (TBGDiff) is proposed to address VSD. The proposed method presents the dual scale aggregation and shadow boundary aware attention modules. Good detection performance is achieved by introducing space-time encoded embedding to guide the diffusion model.

**Strengths:**

1. The use of diffusion models to address VSD is novel.
2. The experiment results on the ViSha dataset are promising.

**Limitations:**

1. In Section 1, it is important to provide a more detailed and clear articulation of the motivation behind the proposed method (e.g., advantages and benefits of using the Diffusion model for video shadow detection) and the reasons for its effectiveness. This will help readers better understand the rationale for the work.

2. In Section 4.1, the AdamW optimizer is not cited.

3. Notations.
1) In Eq. (9), the transpose symbol could not be represented using ‘T’ because it has already been defined as a frame sequence number in the previous sections.
2) In L273-L274, ‘C’ denotes the channels. Also, ‘C’ is the number of the channel to scale in Eq. (9). Are these two C’s the same?
3) In Eq. (1), ‘M’ denotes a function. However, the description above says that ‘M’ is an affinity matrix. Please use different symbols for this.
4) In Fig. 1, what does the D in the green block indicate? If it is a decoder, what is its exact structure?

4. Typos
1) In L112, ‘(3) we explore’ -> ‘(3) We explore’.
2) In L274, ‘the C denote’ -> ‘the C denotes’.
3) In L915, ‘our TBGiff’ -> ‘our TBGDiff’.

**Suitability:**

3

---

### Official Review · Reviewer_p1HA · 2024-05-21

**Rating:** 5
**Confidence:** 3

**Summary:**

The paper introduces a diffusion-based method for video shadow detection. The key ideas include leveraging cross-frame temporal guidance and edge information to improve shadow detection. The method is consists of three modules. A dual scale aggregation module is designed for better temporal understanding, while a shadow boundary aware attention mechanism captures shadow characteristics. The usage of space-time encoded embedding enhances temporal guidance for shadow detection. Experimental results demonstrate that TBGDiff outperforms state-of-the-art methods in both performance, model size, and speed.

**Strengths:**

1.	The paper introduce the first diffusion-based method for VSD.
2.	Extensive experiments demonstrate its effectiveness.
3.	Ablation studies verify that each component is an optimal solution.

**Limitations:**

1.	There is some writing issues may hinder audience from understanding. 1) Table 3 should not list the column of diffusion, if you have not conduct ablation study for it. 2) The name of PCE and PEE is confusing. The actual difference between them and STEE is directional (causal). 3) Auxiliary head and loss take respond for DSA, but it appears at Sec 3.3 SBAA.
2.	The implementation details of diffusion model and guidance encoder are not clear. What lightweight guidance encoder is used?
3.	There are a lot of boundary-aware methods[1,2,3,4] developed in dense-prediction tasks, such as salient object detection, parsing, segmentation, etc. The authors have not discussion the difference between the proposed method and existing ones on handling boundary information. Considering the rationale of the proposed methodology, it should be contained in the introduction and ablation study for clarity.

[1] InverseForm: A Loss Function for Structured Boundary-Aware Segmentation, CVPR21

[2] PlaneSeg: Building a Plug-In for Boosting Planar Region Segmentation, TNNLS23

[3] Decoupled multi-task learning with cyclical self-regulation for face parsing, CVPR22

[4] Synthesize Boundaries: A Boundary-aware Self-consistent Framework for Weakly Supervised Salient Object Detection, TMM23

**Suitability:**

3

---

### Official Review · Reviewer_xbYH · 2024-05-21

**Rating:** 5
**Confidence:** 2

**Summary:**

The paper presents a Time and Boundary Guidance Diffusion (TBGDiff) network for video shadow detection. TBGDiff integrates past-future temporal guidance and boundary information, designs a Dual Scale Aggregation (DSA) module for better understanding of temporal information, and proposes an exploration of Spatial-Temporal Encoding Embedding (STEE) to inject temporal guidance into the diffusion model. For boundary information, it introduces Shadow Boundary Aware Attention (SBAA) to capture shadow characteristics.

**Strengths:**

1. The proposed model TBGDiff effectively combines past-future temporal guidance with boundary information, providing a robust solution for video shadow detection.
2. The DSA module is well-designed and capable of handling both short-term and long-term temporal information, enhancing the model's comprehension of temporal context.
3. The SBAA module incorporates boundaries and masks as auxiliary information and combining them with the attention mechanism, it captures shadow boundary characteristics, strengthening the model's ability to distinguish between shadow and non-shadow areas.
4. The STEE  module allows for parallel guidance encoding and incorporation of temporal information across all timelines by obtaining masks in advance, which is efficient and performs exceptionally well.
5. The paper conducted comprehensive experiments to demonstrate the superior performance of the proposed method.

**Limitations:**

1.  Although the introduction of the diffusion model in video shadow detection is innovative, other components such as the Dual-Scale Aggregation (DSA) and Shadow Boundary Awareness Attention (SBAA) do not show significant innovation based on existing concepts.
2. Regarding the methods proposed in the paper, such as DSA and SBAA, what is the impact on GPU memory usage when computing long sequences simultaneously?

**Suitability:**

2

---

### Official Review · Reviewer_6RQL · 2024-05-26

**Rating:** 4
**Confidence:** 3

**Summary:**

This paper proposes a timeline and boundary guided diffusion network for the task of video object detection. The work successfully embeds the diffusion model into the video shadow detection network and achieves SOTA performance.

**Strengths:**

1. The paper is clear and well-organized.
2. The experiment shows a strong performance.

**Limitations:**

1. Although DSA can capture temporal information from both short and long term sequences. What is the exact parameter of short and long terms? Then the table. 4 shows that DSA with short temporal scales obtains worse performances than the model without DSA, why does the short temporal information weakens the proposed model?
2. It seems that the model predicts the shadow mask of a frame using both past and future frames, does this strategy make the last frames of a video unpredictable and how to solve this problem?
3. Self-attention and diffusion models are both time-consuming modules, and the table.2 shows the model has higher fps than the previous works, so how does the author improve the speed of the model?

**Suitability:**

2

---

### Meta-Review · Area_Chair_XgDn · 2024-07-03

**Recommendation:** Accept (Oral)
**Confidence:** 5

**Metareview:**

A timeline and boundary guided diffusion network has been proposed for the task of video shadow detection. Temporal information is considered through a Dual Scale Aggregation (DSA) and a Spatial-Temporal Encoding Embedding (STEE), while boundary information is captured through a Shadow Boundary Aware Attention (SBAA).

The proposed method is novel, being the first to apply diffusion model for shadow detection. Exploring the temporal guidance for the diffusion model to conduct shadow detection is also interesting. The experiments are solid, and the results are promising.

Most of the concerns previously raised by reviewers have been addressed in the rebuttal. The authors should also incorporate these changes into the paper upon acceptance.